# The Tissue Expression Divergence of the WUSCHEL-Related Homeobox Gene Family in the Evolution of Nelumbo

**DOI:** 10.3390/plants14131909

**Published:** 2025-06-21

**Authors:** Juanjuan Li, Yue Zhang

**Affiliations:** 1Hubei Province Research Center of Engineering Technology for Utilization of Botanical Functional Ingredients, Hubei Key Laboratory of Resource Utilization and Quality Control of Characteristic Crops, College of Life Science and Technology, Hubei Engineering University, Xiaogan 432000, China; juanjuan_black@hbeu.edu.cn; 2Aquatic Plant Research Center, Wuhan Botanical Garden, Chinese Academy of Sciences, Wuhan 430074, China; 3State Key Laboratory of Plant Diversity and Specialty Crops, Wuhan Botanical Garden, Chinese Academy of Sciences, Wuhan 430074, China

**Keywords:** WOX, evolution, expression pattern, co-expression network, Nelumbo

## Abstract

The yellow flower lotus (*Nelumbo lutea*) is the sister species of the sacred lotus (*N. nucifera*). The evolution of gene expression patterns across multiple tissues during the species divergence of these two lotuses remains unexplored. The WUSCHEL-related homeobox (*WOX*) family, a plant-specific transcription factor family, plays a crucial role in tissue development and stress responses. In this study, utilizing a chromosome-level reference genome and a transcriptome database covering multiple tissues, we identified and categorized 11 *NlWOX* genes into three subfamilies. We identified seven syntenic gene pairs between *NnWOX*s and *NlWOX*s that originated from whole-genome duplications. Through conserved motif analysis, we found subfamily-specific motifs in the protein sequences of *NnWOX*s and *NlWOX*s. Variations in the three-dimensional conformations of homologous *WOX* genes indicate function divergences between the two lotus species. The gene expression matrix of *NlWOX* across tissues reveals expression divergences within *N. lutea* and between the two lotus species. By employing a weight gene co-expression network analysis pipeline, we developed eight *NlWOX* co-expression networks that differed from the co-expression networks of their syntenic genes. Overall, our findings suggest that genomic variations in the *WOX* orthologs contribute to the distinct expression patterns and regulatory networks observed during the evolution of these two lotuses.

## 1. Introduction

Transcription factors (TFs) regulate gene expression by binding to cis-acting elements in the upstream promoter region, thereby playing vital roles in various plant growth and development pathways [1,2]. Comparative genomics typically examines changes in appearance or loss within different transcription factor families [3]. For instance, the evolution of transcription factor families involved in drought tolerance and phytohormone response between the early land plant *Physcomitrella patens* and *Arabidopsis thaliana* highlights their significance in the emergence of land plants [4]. However, the molecular functions of TFs are closely linked to specific cellular pathways and morphological features [5,6,7], necessitating further investigations at various levels [8]. Conserved regulatory relationships between TFs and their targeted genes exhibit similar compositions in gene co-expression networks across diverse species [9,10]. The evolution dynamics of TFs at the expression level have garnered considerable attention in research.

*WUSCHEL*-related homeobox (*WOX*) genes are part of a large family of plant-specific transcription factors containing a homeodomain (HD). The main identifying feature of the *WOX* gene is a conserved helix-loop-helix-turn-helix structure consisting of 60 amino acids [11]. Utilizing extensive transcriptomic and protein–protein interaction data from the model plant *Arabidopsis*, both forward and reverse genetic studies on 15 identified *Arabidopsis WOX* (*AtWOX*) proteins have highlighted their roles in organ formation and tissue development, including functions in the shoot apical meristem [12,13], ovule development [14], and floral transition [15]. Phylogenetic analyses have categorized *AtWOX* genes into three subfamilies: the ancient clade, the intermediate clade, and the WUS clade [16,17]. Recent investigations into the biological functions of *WOX* members have revealed their involvement in responses to abiotic stresses and transcriptional regulation [18,19,20]. For instance, in tomatoes, *WOX* gene expression variations were linked to responses to cold, salt, and drought stresses [21]. In rice, the overexpression of *OsWOX11* enhanced tolerance to potassium scarcity and promoted root growth [22]. Additionally, the complex formed by *OsWOX3B* and *OsSPL10* regulates the transcription of *HL6*, which is essential for trichome formation [23]. Genome-wide identifications of WOX genes have been conducted in various species [24,25], and distinct regulatory interactions of WOX orthologs exhibited similar responses, such as *PagWOX11/12a*-*SAUR36* in poplar [26], *OsWOX13*-*OsDREB1A*/*1F* in rice [27], and *MdWOX13-1*-MdSOD in Rosaceae [19]. These findings suggest that the diverse interactions of *WOX* orthologs across species may represent different components in a molecular network governing the development of specific traits. With the widespread use of RNA-seq, gene co-expression network analysis is commonly used to cluster genes with similar expression patterns into modules [28]. Establishing connections between TFs and their co-expressed genes is effective in identifying the downstream-regulated genes of TFs, as genes within a module often participate in similar biological pathways [29]. Nevertheless, the evolution of the co-expression networks of *WOX* genes during plant species divergence remains largely unknown.

Following the last ice age, the Nelumbonaceae family currently comprises only two extant sister species globally: *Nelumbo nucifera* and *N. lutea*. A notable distinction in tissue morphology between these two species lies in their flower color, as *N. nucifera* exhibits flowers ranging from white to red, while *N. lutea* lotus flowers are yellow [30]. *N. nucifera*, commonly known as the sacred lotus, serves as a popular aquatic vegetable in East Asia, valued for its edible and medicinal properties. Consequently, extensive genomic studies on various lotus tissues have generated a substantial amount of genomic data [31,32], which is archived in the Nelumbo Genome Database [33]. Leveraging this resource, our prior research identified the complete set of *NnWOX* genes in *N. nucifera* and discerned their expression variations across diverse tissues [34]. Recently, a high-quality assembly of *N. lutea* and RNA-seq from multiple tissues at various developmental stages became available [35,36]. This development allows for a comparative analysis of gene co-expression networks in these closely related species, offering a novel perspective on the evolutionary trajectory of *WOX* genes. In this investigation, we characterized the *WOX* family members in *N. lutea*, examining their physicochemical attributes, genomic positions, and expression patterns across different tissues. Through a comparative assessment of the co-expression networks of *WOX* genes in *N. nucifera* and *N. lutea*, we observed relatively consistent evolutionary patterns, suggesting that orthologous *WOX* genes likely play analogous regulatory roles in both species.

## 2. Results

### 2.1. Identification of WOXs in N. lutea

Genome assembly and gene annotation of *N. lutea* were downloaded from the Nelumbo Genome Database (http://nelumbo.cngb.org/nelumbo/, accessed on 17 June 2025), and we further mapped the protein sequences to the orthologous public database to annotate their biological functions. To obtain the comprehensive *WOX* gene family members in *N. lutea*, the *WOX* protein sequences of the model plant *A. thaliana* and the basal angiosperm *Amborella trichopoda* were downloaded as the query sequences. We compared the sequence ortholog between the query *WOX* sequences and the protein of *N. lutea* using BlastP. According to the function annotation and orthologous mapped results, we filtered a candidate group of *N. lutea WOX* proteins (*NlWOXs*). We further eliminated candidate proteins without typical conserved domains and redundancies and identified 11 *WOX* gene family members in *N. lutea.* These include *NL1g_04069*, *NL1g_04550*, *NL1g_06482*, *NL2g_10541*, *NL2g_11918*, *NL2g_11942*, *NL2g_12810*, *NL4g_22694*, *NL5g_27723*, *NL5g_28623*, and *NL6g_29420.* Unlike the sequential nomenclature of WOXs in other species based on the location of chromosomes, we named the *NlWOXs* according to their orthologs in *A. thaliana*. As the nomenclature of *NnWOXs* is also based on their orthologs in *A. thaliana*, it is convenient to compare the orthologs between species. Finally, 11 *NlWOX* genes were obtained, including *NlWOX1*, *NlWOX2*, *NlWOX3*, *NlWOX4*, *NlWOX5a*, *NlWOX5b*, *NlWOX9*, *NlWOX11*, *NlWOX13a*, *NlWOX13b*, and *NlWUS* (Table 1). Compared to the number of *WOX* genes in *N. nucifera*, the *WOX* gene family is contracted.

The number of amino acids for the *NlWOX*s ranges from 124 to 362, with an average length of 244.72. Meanwhile, in the *NlWOX*s, the maximum molecular weight is 39945.94 kDa, and the minimum molecular weight is 14769.82 kDa (Table 1). The length of the longest WOX protein in *N. nucifera* and *N. lutea* is similar and is the ortholog of WOX9 in *A. thaliana*. The shortest WOX protein in *N. lutea* is *NlWOX3,* and in *N. nucifera*, it is *NnWOX5*, suggesting the sequence evolution of WOX members in Nelumbo. Similarly, we analyzed the chemical and physical characteristics of NlWOX proteins, including the theoretical PI values, instability index, aliphatic index, and grand average of hydropathicity (Table 1). Most *NlWOX* proteins (10/11) were predicted to have at least one N-glycosylation site (Appendix A). *NlWOX1* has the most, with four N-glycosylation sites, followed by *NlWOX9,* which has three N-glycosylation sites. Interestingly, *NlWOX13a* was identified to have no N-glycosylation sites on the sequence, but its paralogous *NlWOX13b* was predicted to have one, suggesting the sequence evolution in *NlWOX* genes. In addition, we analyzed the hydrophobicity and hydrophilicity for each *NlWOX* protein (Appendix A).

### 2.2. Phylogenetic Tree of NlWOX Proteins

Previous studies suggest that lotus is one of the basal angiosperms and occupies an important position in phylogenetic evolution. To explore the evolutionary relationships of *WOX* genes in lotus and other plants, phylogenetic analysis was performed based on the sequences from *N. lutea* (11), *N. nucifera* (15), *A. thaliana* (15), and *Amborella trichopoda* (9) (Figure 1). In line with previous studies [37], the *WOX* genes across various species were categorized into three clades: ancient clade (AC), intermediate clade (IC), and WUS clade (WC) (Figure 1). Seven *NlWOX* genes were divided into WC clades, and the other two clades had two *NlWOX* genes, respectively. We found the *AmtrWOX* gene in each small branch, suggesting that these *WOX* genes originate from basal angiosperms. In comparison to the *WOX* genes in *A. thaliana* or *A. trichopoda*, *NlWOX*s show a closer evolutionary relationship to *NnWOX*s. This suggests that these two lotus species share a similar evolution in the *WOX* gene family. Notably, we found that *NlWOX13a* is closer to *NnWOX13a* rather than *NlWOX13b*, indicating a stronger sequence variation in paralogs of *N. lutea* than in orthologs of the two lotus species.

### 2.3. Gene Duplication and Synteny Analysis

To investigate the genetic distances of *NlWOX* genes, the chromosomal positions of 11 *NlWOX* genes are exhibited in Figure 2a. In terms of eight pseudochromosomes, the *NlWOX* genes are distributed in only five pseudochromosomes (i.e., chr1, chr2, chr4, chr5, and chr6). We found four *NlWOX* genes (*NlWOX2*, *NlWOX3a*, *NlWOX5a*, and *NlWOX5b*) located on chr2, indicating their close genetic distances. To further explore whether the physical bunching of *NlWOX* genes in chr2 was caused by whole-genome duplications, five different gene duplication types were identified using MCScanX. Only two duplicated gene pairs were found in the whole-genome duplication block of paralogs in *N. lutea*. The *NlWOX5a*/*NlWOX5b* duplicated gene pair was in the same chromosome (chr2), while the *NlWOX13a*/*NlWOX13b* duplicated genes were located in different chromosomes (Figure 2b). Our results indicate that whole-genome duplication events on chr2 may have contributed to the largest number of *NlWOX* genes. However, most *NlWOX* genes were not copied along with the whole-genome duplication or lost their duplications during evolution.

Orthologous genes change their chromosomal locations during the formation of species. To study the evolution of *WOX* genes on the genome during the divergence of two lotus species, an interspecies collinearity analysis of orthologous genes was carried out between *N. nucifera* and *N. lutea*. Based on the synteny regions, we identified a total of 13 collinearity gene pairs of *WOX* (Figure 2c). We found seven conserved syntenic gene pairs, including *NlWOX13a*-*NnWOX13a*, *NlWOX1*-*NnWOX6b*, *NlWOX9*-*NnWOX9b*, *NlWOX4*-*NnWOX4a*, *NlWOX5a*-*NnWOX5b*, *NlWOX3*-*NnWOX3a*, and *NlWOX5b*-*NnWOX5a*. Notably, three duplicated gene pairs, *NnWOX3a*/*3b*, *NnWOX4a*/4*b*, and *NnWOX6a*/*6b*, in *N. nucifera* were uniquely syntenic to *NlWOX3a*, *NlWOX4*, and *NlWOX1*, respectively. This indicates the direction of whole-genome duplication events in the *N. nucifera* genome, i.e., *NnWOX3b* was duplicated from *NnWOX3a*, *NnWOX4b* was duplicated from *NnWOX4a*, and *NnWOX6a* was from *NnWOX6b*. On the contrary, the duplicated genes of *NlWOX3a*, *NlWOX4*, and *NlWOX1* might be lost after whole-genome duplication. In addition, no synteny genes were identified for *NlWUS*, *NlWXO2*, *NlWOX11*, and *NlWOX13b*, indicating their unique evolution of genome sequences.

### 2.4. Conserved Motifs and Protein Conformation of WOX Genes in Nelumbo

To explore the sequence evolution of syntenic and non-syntenic *WOX* genes between two lotus species, the conserved motifs were predicted in both *NlWOX* and *NnWOX* proteins. For all *WOX* proteins in the lotuses, Motif1 and Motif2 were identified but in different locations of *WOX* proteins (Figure 3a). In the WUS subfamily, except for *NlWOX3*, *NlWOX2*, and *NnWOX2*, *NlWOX* and *NnWOX* proteins contained Motif8 (Figure 3a). Motif10 was specifically identified in *NlWUS* and *NnWUS* (Figure 3a). In the AC subfamily, only *NlWOX13b* was found to lose Motif5, and all *WOX* proteins contained the subfamily-specific Motif7 (Figure 3a). Motif4 was identified to be IC subfamily-specific (Figure 3a). Notably, we found that syntenic *WOX* gene pairs show a high similarity of conserved motifs, suggesting that they might have the same functions in lotuses.

Based on the arrangement of amino acids in the polypeptide chain, the secondary structures of *NnWOX* and *NlWOX* proteins were predicted (Figure 3b). Random coils are the predominant secondary structure in the *WOX* protein sequences, accounting for 62.31% to 83.88% of all protein sequences (Figure 3b). Interestingly, alpha helices are more prevalent in the sequences of *NlWOX3*, *NlWOX13a*, and *NnWOX13a* (Figure 3b). Furthermore, a three-dimensional structure analysis of *NnWOX* and *NlWOX* proteins was performed (Figure 3c). The conserved helix-loop-helix-turn-helix structure was identified in the three-dimensional conformation of *NnWOX* and *NlWOX* proteins (Figure 3c). We found specific three-dimensional structures at the end of *NlWOX9*, *NlWOX11*, *NlWOX13a*, *NnWOX9a/9b*, *NnWOX11*, and *NnWOX13a/13b*, indicating that they may be involved in a specific protein complex. Most of the syntenic genes showed similar three-dimensional structures. However, *NlWOX3* exhibited more loose protein conformation than its syntenic *NnWOX3a*, suggesting weaker noncovalent interactions.

### 2.5. Tissue Expression Pattern Divergence of WOX Proteins in Lotuses

Previous studies indicate that *WOX* genes are widely expressed in multiple tissues in the model plant *Arabidopsis* [38] and rice [39]. Meanwhile, our study found the expression divergence of *NnWOX* genes among different tissues [34]. To further compare the expression patterns of *WOX* genes in two lotuses, the RNA-seq datasets of different tissues in *N. lutea* were downloaded. A transcriptome analysis pipeline was carried out to construct the *NlWOX* gene expression matrix across 36 tissue samples (Figure 4a). Our results show the specific expression pattern of *NlWOX1* in the carpel (Figure 4a), while the orthologous *NnWOX6b* was highly expressed in the carpel and cotyledon [34], suggesting the expression divergence of orthologous gene pairs between the two lotus species. *NlWOX3* was specifically expressed in the apical meristem (Figure 4a), but the syntenic *NnWOX3a* had a wider tissue expression pattern. This highlights the potential role of *NlWOX3* in the development of the apical meristem, which is associated with the asexual propagation of rhizomes in *N. lutea*. *NlWOX4* is not only highly expressed in the apical meristem and carpel, like its orthologous gene *NnWOX4*, but also in the leaf, receptacle, and petal, indicating that the evolution of the tissue expression pattern might result in neofunctionalization between orthologous gene pairs. The *NnWOX5a/5b* were root-specifically expressed, but their orthologous *NlWOX5b/5a* were silent (FPKM < 1) in *N. lutea* tissues (Figure 4a). We speculate that *NlWOX5a/5b* lost its biological function or was downregulated by post-transcriptional regulatory mechanisms. We found that *NlWOX13b* was highly expressed in most tissue samples but is downregulated in the cotyledon at 12 and 15 days after pollination. Interestingly, the paralog *NlWOX13b* is only highly expressed in certain tissues, including the apical meristem, rhizome node, seed coat, radicle, internode, and petiole. This suggests that the complementary tissue expression patterns of these two duplicated genes might result from the biological function redundancy and subfunctionalization after duplication. Compared to the widely high expression levels of *NnWOX13a* across different tissues [34], the orthologous *NlWOX13a* was specifically expressed in certain tissues, indicating that the syntenic gene pairs underwent different functional divergences. We also found the conserved tissue expression patterns of *NlWOX9* and *NnWOX9b*; therefore, these two orthologous genes from the AC subfamily continued the same biological function during evolution.

Since the transcription of the gene is activated by the trans-acting factors that bind the cis-regulatory elements, we further predicted the cis-regulatory elements in the promoter regions of *NlWOX* and *NnWOX* genes (Figure 4b). We found that cis-regulatory elements were different between the duplicated genes in both *N. nucifera* and *N. lutea*, suggesting that genetic variations in the promoter regions might result in the expression divergence of paralogous genes (Figure 4b). Similarly, both the type and number of cis-regulatory elements between the promoter regions of orthologous gene pairs were distinct (Figure 4b). Our results indicate that the tissue expression patterns of *WOX* genes are highly associated with the cis-regulatory elements that affect the transcriptional activity.

### 2.6. qRT-PCR Experiments of NlWOX Genes

To verify the tissue expression patterns of the *NlWOX* genes, we first collected five *N. lutea* tissue samples. According to the expression matrix of *NlWOX* genes, we selected six expressed genes in the collected tissues to perform the qRT-PCR experiments. Our results suggest that *NlWOX1*, *NlWOX3*, and *NlWOX4* showed significantly (ANOVA test, *p*-value < 0.01) higher relative expression levels in the apical meristem than in other tissues (Figure 5a–c). *NlWOX11* and *NlWOX13b* have significantly higher relative expression levels in the root (Figure 5d,f). The duplicated *NlWOX13a* showed higher relative expression levels in the leaf and petiole, which is different from its duplicates (Figure 5e,f). This indicates the expression divergences between duplicates. In addition, the qRT-PCR results were consistent with the RNA-seq sequencing, demonstrating the accuracy of our results.

### 2.7. Co-Expressed Relationships of NlWOX Genes

Transcription factors can regulate multiple downstream genes that play important roles in plant growth and development. Gene co-expression network analysis is an efficient method to identify gene clusters in the same biological pathway. Given the fact that the weight gene co-expression network analysis (WGCNA) pipeline is widely performed to construct the gene co-expression networks, we first filtered out the silent genes from the gene expression matrix in *N. lutea*. A total of 31,264 expressed genes were input into the WGCNA pipeline and aggregated into fourteen color modules (Appendix A). Based on the WGCNA results, these gene modules were significantly (*p*-value < 0.01) related to unique tissues (Appendix A). Only seven *NlWOX* genes were clustered into four color modules, including *NlWOX1*/*3*/*4* (MEturquois), *NlWOX9*/*11* (MEpink), *NlWOX13a* (MEgrey), and *NlWOX13b* (MEgrey60) (Figure 6a, Appendix A). The MEturquois is significantly associated with the carpel, MEpink is significantly associated with the cotyledon, MEgrey is significantly associated with the rhizome node, and MEgrey60 is significantly associated with the root (Appendix A).

Our previous study constructed the co-expression networks for *NnWOX* genes [34], and we further compared the tissue bias of syntenic *WOX* gene pairs (Appendix A). Due to the similar expression patterns of *NnWOX3a*/*4a*/*6b*, these three *NnWOX* genes from the WC subfamily were identified in one gene module that is significantly related to the cotyledon and apical meristem, whereas their syntenic *NlWOX1*/*3*/*4* genes are clustered into the carpel-specific MEturquois, suggesting that the syntenic genes might be involved in different tissue developments by constructing distinct co-expression networks (Figure 6a). Similarly, *NnWOX13a* was in the gene module associated with the seed coat, but its syntenic *NlWOX13a* was significantly related to the rhizome node. However, the syntenic gene pair *NnWOX9b*-*NlWOX9* is significant in the same tissue, the cotyledon, indicating its conserved expression pattern and co-expression relationship. Furthermore, a GO enrichment analysis of the co-expressed genes for the *NlWOXs* was performed (Figure 6b). The top three significantly enriched GO terms in the biological process are “DNA metabolic process”, “cell cycle”, and “cellular component organization”. Notably, the most enriched GO terms between the co-expressed genes for *NnWOXs* and *NlWOXs* were distinct, suggesting the *WOX* genes participated in different biological pathways between the two lotus species.

## 3. Discussion

Plant genome assemblies have facilitated the study of gene family evolution, encompassing expansion or shrinking across various species. The *WOX* gene family is a highly conserved and plant-specific transcription factor family that plays a pivotal role in governing tissue development and responses to abiotic stresses [40,41,42,43]. Recognizing the significance of the *WOX* family, our previous study carried out genome-wide analysis in *N. nucifera*, resulting in the identification of 15 *NnWOX* genes. Aimed at delving deeper into the evolution of *WOX* in Nelumbo, a total of 11 *NlWOX* genes were identified in *N. lutea*, the only sister species to *N. nucifera*. Bioinformatic assessments of the *NlWOX* genes, including phylogenetic trees, conserved motifs, the physicochemical properties of proteins, protein structures, cis-regulatory elements, and tissue expression patterns, provide a comprehensive framework for molecular experiments in *N. lutea*.

Being one of the basal angiosperms, the lotus species underwent a single ancient whole-genome duplication (WGD) event, as evidenced in the *N. nucifera* genome [31]. Changes in the lotus gene family composition, whether through gains or losses, are linked to their adaptive responses to environmental shifts. The comparatively lower number of *WOX* genes in *N. lutea*, in contrast to *N. nucifera*, hints at distinct evolutionary trajectories of genomes in these two lotus species. In this investigation, building upon the identified *NnWOXs*, we adopted nomenclature for lotus *WOX* genes based on homologous genes from the model plant *A. thaliana*, diverging from the chromosome position-based naming convention in other species [44]. Despite variations in the size of the *WOX* gene family, all genes consistently segregate into three subfamilies [45,46].

Within the AC subfamily, the *WOX* genes of both lotus species underwent a whole-genome duplication event. Notably, only the gene pair *Nl*/*NnWOX13a* remains collinear, suggesting that the genomic region near *Nl*/*NnWOX13b* underwent mutations after the genome duplication. This led to its loss of collinearity. Remarkably, the lotus lacks an orthologous gene to *AtWOX14*, which regulates the development and cell differentiation in the vascular bundle [47,48], aligning with the hollow vascular tissue in the lotus leaf petiole utilized for gas transport [49]. In the IC subfamily, a solitary duplication event occurred in *N. nucifera* (*NnWOX9a*/*9b*). While the WC subfamily stands as the largest within the *WOX* gene family, only one duplicated gene pair was identified in *N. lutea*, contrasting with three pairs in *N. nucifera* (Figure 1). In Boehmeria nivea, five duplication events expanded the *WOX* gene family [50]. We posit that differences in whole-genome duplication events and various duplication types contribute to *WOX* gene family expansion, with functional redundancy post-genome duplication potentially leading to gene losses during evolution. Notably, *WOX5a*/*b* exhibits collinearity between the two genomes, indicating they underwent a shared ancient duplication event. Compared to *N. nucifera*, *N. lutea* has fewer duplicated *WOX* genes, suggesting that *N. lutea* may have experienced *WOX* gene loss after whole-genome duplication. Therefore, further comparative genomics investigations on WGD in the two lotus species are essential to clarify the relationship between WGD events and the gain or loss of *WOX* gene family members.

Due to the hypothetical surface of a protein consisting of the regions defined by various three-dimensional structures detected by amino acid sequences [51], the protein sequences of homologous *WOX* genes would have distinct folded states and functional regions. Sequence motif results indicate that *WOX* genes in two lotus species have two conserved motifs (Motif1 and Motif2). The AC and IC subfamily members contained unique motifs, such as Motif7 in AC and Motif4 in IC. Subfamily-specific motifs were also identified in other plants [52], suggesting the conserved motifs in the same subfamily. We found that the percentage of protein secondary structures in paralogous and syntenic genes was also different (Figure 3b), revealing the genomic variations contributing to the changes in protein structures. Notably, we predicted the three-dimensional structures for both *NlWOX* and *NnWOX* genes, highlighting the same conformation of the conserved homeodomain. The three-dimensional conformations between duplicated genes of different duplication types are different. This result indicates that all duplicated genes undergo neofunctionalization during the evolutionary process. Otherwise, they would be lost due to functional redundancy. Differentiation in the three-dimensional conformations between the syntenic gene pairs (*NlWOX1*-*NnWOX6b*, *NlWOX9*-*NnWOX9b*, and *NlWOX3*-*NnWOX3a*) has been observed, especially the very loose conformation of *NlWOX3*. However, a high similarity in the three-dimensional conformations between the syntenic gene pairs (*NlWOX4*-*NnWOX4a*, *NlWOX5a*-*NnWOX5b*, *NlWOX5b*-*NnWOX5a*, and *NlWOX13a*-*NnWOX13a*) has also been found. This suggests that the two genomes of the lotus have undergone variation in specific regions while also retaining some conserved collinear regions.

Transcriptional factors contribute to certain cellular pathways or morphological features in the context of downstream targeted genes [53]. However, how TF expression patterns and regulatory networks evolved during the divergence of plant species is largely unclear. We compared the tissue expression patterns and co-expression networks of syntenic *WOX* gene pairs. Notably, we found the significant expression divergence of syntenic genes across different tissues at different developmental stages, indicating their distinct roles in regulating the phenotypes of lotuses (Figure 4a). Recently, studies suggested that *WOX13* promotes cellular reprogramming and organ regeneration involved in various plant tissues [54,55]. *NnWOX13a*/*13b* exhibited consistently high expression levels in multiple tissues. However, *NlWOX13a*/*13b* has complementary high expression patterns, i.e., *NlWOX13a* is highly expressed in specific tissues where *NlWOX13b* is lowly expressed. Our results indicate that *NnWOX13a*/*13b* might have redundant functions, whereas *NlWOX13a*/*NlWOX13b* has completed the mutually exclusive assignment of ancestor functions or subfunctionalization. Furthermore, we compared the WGCNA co-expression network of syntenic *WOX* genes between *N. nucifera* and *N. lutea*. Syntenic genes were clustered into tissue-specific gene modules, indicating their different regulatory networks. We hypothesize that differences in the promoter regions of these collinear genes may contribute to their tissue specificity. The enriched biological functions of co-expressed genes for *NnWOX*s and *NlWOX*s diverged, suggesting that they participated in different pathways. Therefore, significant divergences of *WOX* genes in both tissue expression patterns and co-expression networks contributed to their distinct roles in regulating the growth and development of tissues, along with the evolution of the lotus species.

## 4. Materials and Methods

### 4.1. Identification of WOX Genes in Nelumbo lutea

The protein database of *N. lutea* was downloaded from the Nelumbo Genome Database [33,35] (http://nelumbo.cngb.org/nelumbo/, accessed on 17 June 2025). Biological function annotations of these proteins were predicted using EggNOG v5.0 [56]. To identify the *WOX* gene family members in *N. lutea*, we first filtered a candidate pool according to the annotation of the WUSCHEL-related homeobox. Meanwhile, a total of fifteen *AtWOX* genes in the model plant *Arabidopsis thaliana* and nine *AmtrWOX* genes in the basal angiosperm *Amborella trichopoda* were retrieved from the Plant Transcription Factor Database (PlantTFDB, https://planttfdb.gao-lab.org/, accessed on 17 June 2025) and further mapped to the protein sequences of *N. lutea* using BlastP. The mapped results were filtered with a *p*-value < 1 × 10^−5^ and a mapped ratio of targeted sequence length >0.8. Previous studies indicated that *WOX* orthologs contained a conserved HB domain, which specifically bound DNA sequences. Consequently, the candidate *NlWOX* genes were mapped to genome databases, including the NCBI Conserved Domain Database (CDD, https://www.ncbi.nlm.nih.gov/Structure/cdd/cdd.shtml, accessed on 17 June 2025), the SMART database (http://smart.embl-heidelberg.de/, accessed on 17 June 2025), and the Pfam database (https://pfam.xfam.org/, accessed on 17 June 2025). Only mapped results that were identified to have the HB domain region, i.e., cl00084 in CDD, SM000389 in SMART, and PF00046 in Pfam, were retained for subsequent analysis. Finally, the identified *NlWOX* genes were named according to their orthologs in *Arabidopsis thaliana*, and duplicates were named alphabetically.

### 4.2. Chromosome Location and Physicochemical Characteristics of NlWOX Genes

Genome sequences and gene locus information of the physical location of *N. lutea* were downloaded from the Nelumbo Genome Database [33,35] (http://nelumbo.cngb.org/nelumbo/, accessed on 17 June 2025). Zhang et al. annotated the genome of *N. lutea* with multiple transcripts in the gene loci; the longest transcript sequence from each gene was used to represent the protein-coding regions. The chromosome position of *NlWOX* genes was drafted using TBtools-II software [57]. To understand the physical and chemical characterizations of the *NlWOX* proteins, we used the Proparam tool of ExPASy (http://weB.expasy.org/protparam/, accessed on 17 June 2025). In the basal feature analysis, we examined several indicators, including the count of amino acids, molecular weights, theoretical PI values, instability index, aliphatic index, and the grand average of hydropathicity. Furthermore, *NlWOX* sequence features were analyzed using NetNGlyc (https://services.healthtech.dtu.dk/services/NetNGlyc-1.0/, accessed on 17 June 2025) to predict the candidate N-glycosylation sites. Protein hydrophobicity and hydrophilicity were estimated using the ProtScale tools of ExPASy (https://web.expasy.org/protscale/, accessed on 17 June 2025). In addition, the sequence motifs of *NlWOX* genes were analyzed using the MEME tools (http://meme-suite.org/meme/tools/meme, accessed on 17 June 2025) with a maximum of ten motifs.

### 4.3. Phylogenetic Analysis of NlWOXs

WOX protein sequences from *N. lutea* (*NlWOX*), *N. nucifera* (*NnWOX*), *A. thaliana* (*AtWOX*), and *A. trichopoda* (*AmtrWOX*) were obtained. We first aligned these *WOX* protein sequences using the muscle. For the aligned sequences, a phylogenetic neighbor-joining tree was constructed using MEGA v7.0, with the parameters “pairwise deletion” and “1000 replicates in bootstrap” [58]. To improve the readability of the phylogenetic tree, iTOL online tools (https://itol.embl.de/, accessed on 17 June 2025) were used to add legends and colors.

### 4.4. Interspecies and Intraspecies Synteny Analyses

As *N. lutea* is the only sister species of *N. nucifera*, we further studied the interspecific synteny of orthologs. The genome sequences and location of genes of *N. nucifera* were also downloaded from the Nelumbo Genome Database. The protein sequences of *N. nucifera* and *N. lutea* were, respectively, extracted according to their gene annotation, and we aligned these proteins using BLASTP with a maximum of six mapped results. The synteny of orthologs between two lotuses was analyzed using MCSanX [59]. We filtered the synteny of *WOX* genes and displayed them with a dual synteny plot using TBtools software [57]. Then, MCSanX was also used to identify the synteny block regions on the *N. lutea* genome. According to the chromosomal distance between duplicated gene pairs, the duplicated types of genes were classified into five groups: singletons, WGD/segmental duplications, dispersed duplications, proximal duplications, and tandem duplications. Further, we identified the duplicated type of *NlWOX* genes and showed them in a circus plot using the RCircos package [60].

### 4.5. GO Enrichment Analysis

Based on the functional annotations of *N. lutea*, the GO enrichment analysis of syntenic genes of *NlWOX*s was carried out using TBtools [57]. The enriched GO terms, i.e., *p*-values < 0.01, are shown in a bar chart.

### 4.6. Tissue Expression Profiles of NlWOXs

To explore the expression divergences of *NlWOX* genes among multiple tissues, the RNA-seq database of 18 tissue samples in *N. lutea* was downloaded from the NCBI SRA dataset under accession number PRJNA705058. To remove the adapter and low-quality reads, the raw reads were filtered using Trimmomatic [61]. Clean reads were mapped to the *N. lutea* reference genome using Hisat2 [62]. Based on the gene loci annotation, the gene expression levels were estimated using the FPKM (fragments per kilobase of exon model per million mapped fragments) value for each tissue sample using StringTie [63]. The tissue expression patterns of *NlWOXs* were extracted from the gene expression profiles. The circlize v0.4.16 R package was used to generate the heatmap circle of NlWOX gene expression. The gene expression profile of 54 tissue RNA-seq samples in *N. nucifrea* was downloaded from the Nelumbo Genome Database.

### 4.7. Quantitative Real-Time PCR Experiments

The *N. lutea* plants were cultivated in the Wuhan Botanical Garden. The apical meristem, internode, leaf, petiole, and root tissues were collected. Total RNA was extracted from each tissue using an RNAprep Plant Kit, and high-quality RNA was reverse-transcribed into cDNA. Based on the expression pattern of *NlWOX* genes from RNA-seq, we filtered six expressed *NlWOX* genes (i.e., *NlWOX1*, *NlWOX3*, *NlWOX4*, *NlWOX11*, *NlWOX13a*, and *NlWOX13b*) in the collected tissues. The specific primers designed for these six genes are shown in Appendix A. qRT-PCR experiments were conducted on the six genes under the following conditions: 95 °C for 30 s; 40 cycles of 95 °C for 5 s; 60 °C for 30 s; and 72 °C for 15 s; and 95 °C for 10 s. The relative expression levels were calculated using the 2^−ΔΔCt^ method.

### 4.8. Comparative Co-Expression Gene Network Analysis

To study the evolution of co-expressed genes in collinear *WOX* genes in Nelumbo, the gene co-expression networks for *N. nucifera* and *N. lutea* were constructed. Briefly, genes that show low expression levels (an average of FPKM in the tissue expression profile of <0.1) were filtered out. Based on the pipeline of weighted gene co-expression network analysis (WGCNA) R-package, we built the WGCNA networks with a min-module of 500 genes using the tissue expression profile in *N. nucifera* and *N. lutea*. The correlations between gene modules and tissues were estimated; the modules that showed a significant (*p*-values < 0.001) correlation to one tissue were defined as tissue-specific modules. We defined the co-expressed genes as the top 5% of genes based on their weight proportions in the co-expression network. The *NlWOX* co-expression networks were extracted and visualized using Cytoscape v3.10 [64].

## 5. Conclusions

This study focuses on the evolution of the *WOX* family in Nelumbo. A total of 11 *NlWOX* genes were identified in *N. lutea* and were classified into three subfamilies. Phylogenetic analysis reveals the evolutionary relationships of lotus *WOX* members with basal angiosperms and model plants. Gene duplication and synteny analyses show that whole-genome duplications affected the distribution and evolution of these genes. Conserved motif and protein conformation analyses indicate variations between the two lotus species and among different subfamilies. In terms of tissue expression patterns, *NlWOX* genes exhibited significant divergence compared to their orthologs in *N. nucifera*, in line with their distinct cis-regulatory elements in promoters. Moreover, the co-expression networks of *NlWOX* genes were distinct from those of *NnWOX* genes, suggesting different regulatory roles in tissue development. Overall, our research provides valuable insights into the evolution and function of the *WOX* family in Nelumbo, highlighting the importance of genomic variations in gene expression and regulatory networks during the evolution of these two lotus species.

## Figures and Tables

**Figure 1 plants-14-01909-f001:**
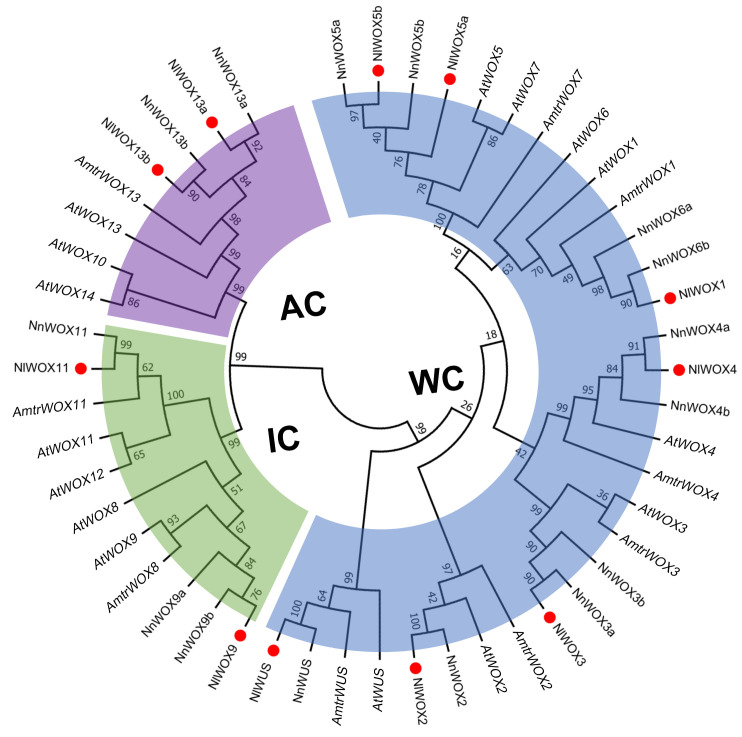
Phylogenetic tree of WOX protein in *N. lutea*, *N. nucifera*, *A. thaliana*, and *A. trichopoda*. Three subfamily clades were identified: ancient clade (AC, purple), intermediate clade (IC, green), and WUS clade (WC, blue).

**Figure 2 plants-14-01909-f002:**
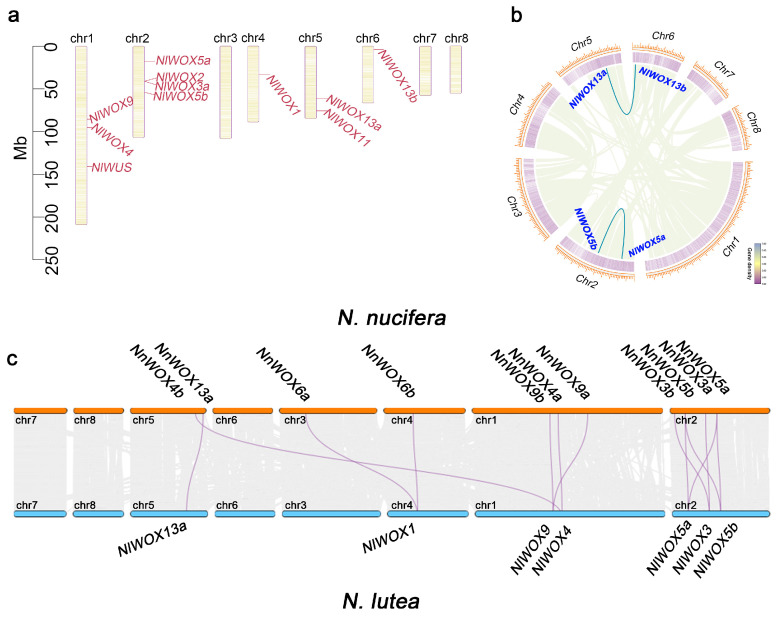
The genome distribution and collinearity relationship of *NlWOX* genes. (**a**) The chromosome location of *NlWOX* genes. The left axis stands for the length of the chromosome. (**b**) Whole-genome duplications of genes in the *N. lutea* genome. The blue lines are the duplicated *NlWOX* genes in the collinearity block. (**c**) The interspecies collinearity gene pairs between *N. lutea* and *N. nucifera*. The WOX genes are highlighted as purple lines.

**Figure 3 plants-14-01909-f003:**
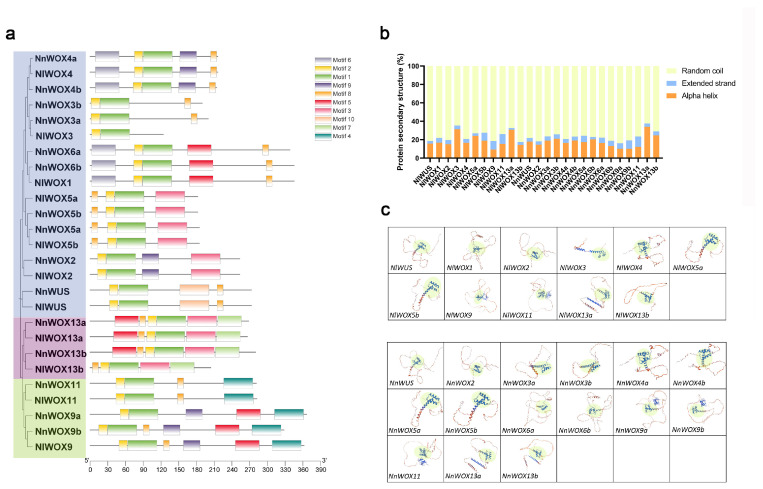
The conserved motif and protein sequence structures of WOX in Nelumbo. (**a**) The conserved motifs of *NlWOX* and *NnWOX* genes were identified and are listed according to the phylogenetic tree on the left. (**b**) The percentage of protein secondary structures in *NlWOX* and *NnWOX* genes. (**c**) The three-dimensional conformations of homologous WOX genes were predicted. The green highlights are the conserved homeodomains.

**Figure 4 plants-14-01909-f004:**
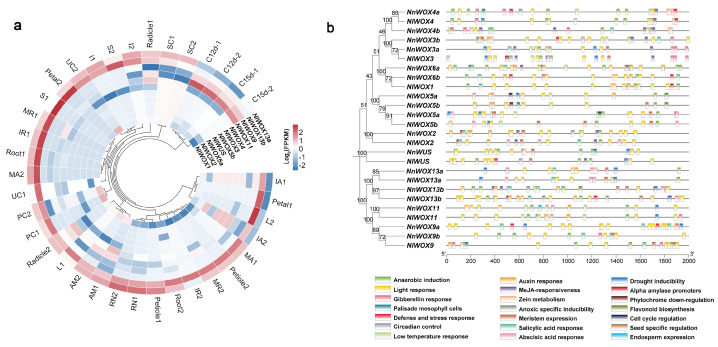
The Tissue expression patterns of *NlWOX*s and cis-regulatory elements in WOX promoters. (**a**) A heatmap circle showing the expression level (log FPKM) of *NlWOX*s across tissue samples. (**b**) The distribution of cis-regulatory elements. Immature anther, IA; mature anther, MA; apical meristem, AM; pollinated carpel, PC; unpollinated carpel, UC; cotyledon-12d, C12d; cotyledon-15d, C15d; internode, I; leaf, L; immature receptacle, IR; mature receptacle, MR; rhizome node, RN; seed coat, SC; sepal, S.

**Figure 5 plants-14-01909-f005:**
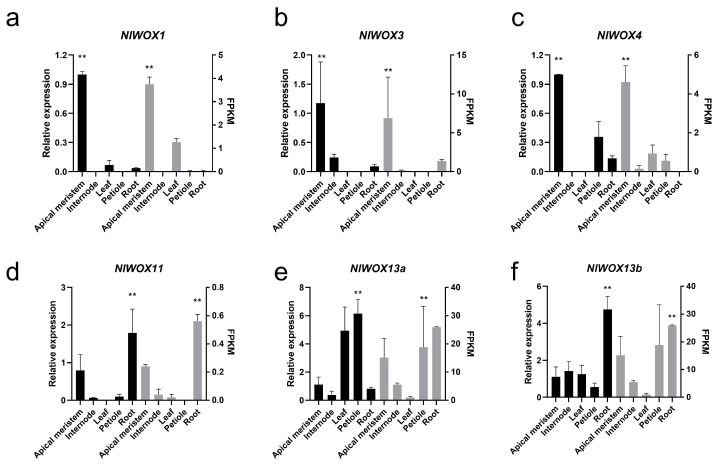
qRT-PCR and RNA-seq show the expression levels of *NlWOX* in five tissues. (**a**–**f**) Six expressed *NlWOX* genes were tested. The black bars show the qRT-PCR results, and the grey bars show the FPKM value. The error bars show the standard error of the mean. Significance was tested by an ANOVA test; ** means *p*-value < 0.01.

**Figure 6 plants-14-01909-f006:**
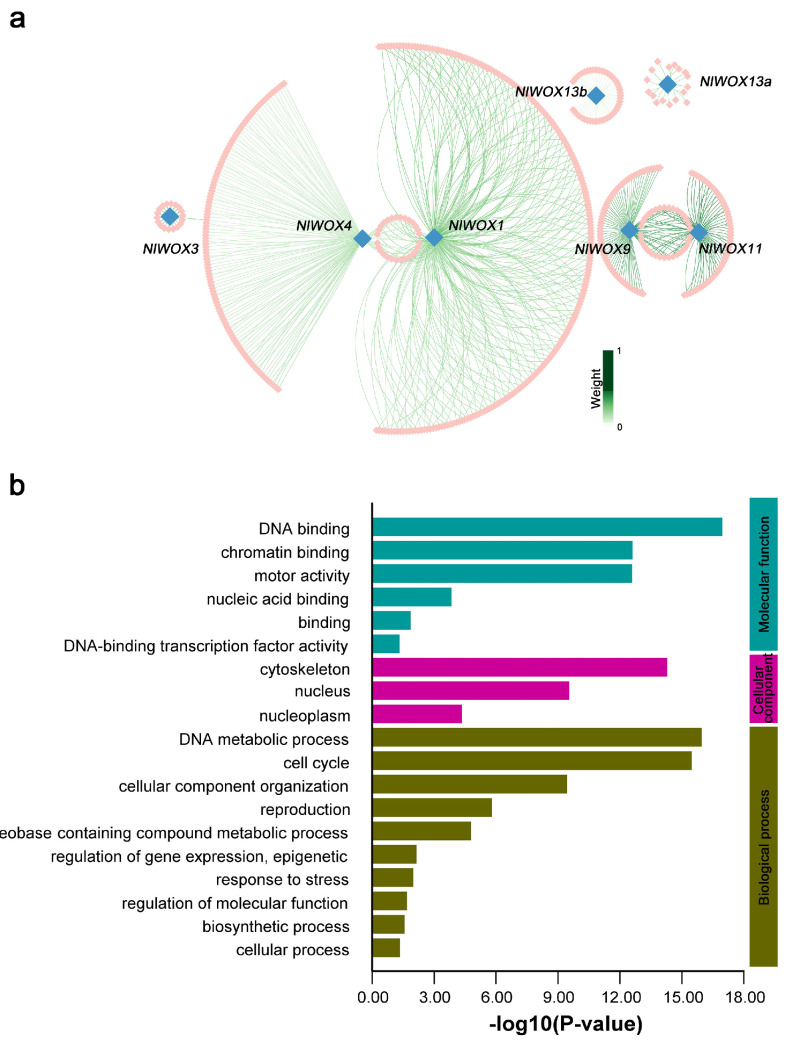
The co-expression network analysis of *NlWOX*s. (**a**) The co-expression networks for each *NlWOX* gene. Blue squares are the *NlWOX* genes, and pink squares are their co-expressed genes. The color in the lines represents their weighted correlation. (**b**) The GO enrichment analysis of *NlWOX* co-expressed genes.

**Table 1 plants-14-01909-t001:** Physicochemical properties of *NlWOX* proteins in *Nelumbo lutea*.

ID	Name	Number of Amino Acids	Molecular Weight	Theoretical PI	Instability Index	Aliphatic Index	Grand Average of Hydropathicity
*NL1g_04069*	NlWOX9	362	39,945.94	7.15	53.16	67.62	−0.464
*NL1g_04550*	NlWOX4	216	24,502.73	9.46	54.77	63.61	−0.946
*NL1g_06482*	NlWUS	273	30,043.36	6.83	63.71	54.32	−0.746
*NL2g_10541*	NlWOX5a	182	20,712.29	6.91	60.57	67.47	−0.795
*NL2g_11918*	NlWOX2	253	27,883.17	6.71	50.01	63.2	−0.611
*NL2g_11942*	NlWOX3	124	14,769.82	10.01	81.83	65.4	−0.99
*NL2g_12810*	NlWOX5b	185	21,006.71	8.7	42.95	68.49	−0.657
*NL4g_22694*	NlWOX1	345	39,317.67	6.43	57.25	53.45	−0.904
*NL5g_27723*	NlWOX13a	266	30,660.55	6.08	58.31	67.82	−0.812
*NL5g_28623*	NlWOX11	282	30,602.15	5.42	75.04	69.79	−0.291
*NL6g_29420*	NlWOX13b	204	22,930.45	6.06	53.57	68.38	−0.793

## Data Availability

All data generated or analyzed during this study are included in this published article.

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
