# Peer review of "The Tissue Expression Divergence of the WUSCHEL-Related Homeobox Gene Family in the Evolution of Nelumbo"

_plants, 2025, doi:10.3390/plants14131909_

Round 1

Reviewer 1 Report

Comments and Suggestions for Authors

Dear Authors,

Congratulations on the paper entitled Genome-wide identification and expression analysis of the WUSCHEL-related homeobox gene family in the evolution of Nelumbo. It is of great importance for vegetal molecular biology. 

However, there are some minor issues to be addressed before publication:

  • Reference 18 - The number is doubled
  • Figures 2, 3, 4 - should have higher resolution 
  • Chen et al. (2023) greatly discussed the NnWOX functions - you should briefly discuss their findings in order to make sure that your article has great value and not redundant information

Author Response

Comment 1: Reference 18 - The number is doubled

Response: We revised this error in reference 18.

Comment 2: Figures 2, 3, 4 - should have higher resolution 

Response: We submit the Figures with higher resolution in additional files.

Comment 3:  Chen et al. (2023) greatly discussed the NnWOX functions - you should briefly discuss their findings in order to make sure that your article has great value and not redundant information

Response: Thank you for your professional comments. I also agree that the study by Chen et al. (2023) is of significant importance for WOX genes in N. nucifera. However, we have previously constructed a co-expression network for NnWOX genes (Li et al., 2024). Considering the comments from the Plants journal: ‘If the reviewer(s) recommended references, critically analyze them to ensure that their inclusion would enhance your manuscript. If you believe these references are unnecessary, you should not include them.’ We believe that citing the literature by Chen et al. (2023) would result in a duplicate citation. Therefore, we did not cite this great paper.

Reviewer 2 Report

Comments and Suggestions for Authors

The manuscript is well written with a nice presentation of data and results.

I have the following comments for the improvement of this manuscript.

Please revise your title since it is very general and does not convey the specific results of your study.

In the results, please revise panel a and b of Figure 2 since they are not readable.

Likewise, Figure 4 is a very important Figure but very difficult to read. Please enlarge the diagrams and the fonts so that Figure 4 is understandable. Figure 4 presents the tissue expression patterns of NlWOXs and cis-regulatory elements in WOX promoters. The Heatmap and distribution of cis-regulatory elements need to be decipherable.

Figure 6 is another important Figure that is difficult to read. Please enlarge and revise accordingly.

The Discussion section needs some revision. There are long sentences and long paragraphs and the text would benefit greatly from some reorganization.

Author Response

Comment 1: Please revise your title since it is very general and does not convey the specific results of your study.

Response: According to your suggestions, we revised our title: “Tissue expression divergence of the WUSCHEL-related homeobox gene family in the evolution of Nelumbo”.

Comment 2: In the results, please revise panel a and b of Figure 2 since they are not readable.

Response: We revised Figure 2 to make it more readable.

Comment 3:  Likewise, Figure 4 is a very important Figure but very difficult to read. Please enlarge the diagrams and the fonts so that Figure 4 is understandable. Figure 4 presents the tissue expression patterns of NlWOXs and cis-regulatory elements in WOX promoters. The Heatmap and distribution of cis-regulatory elements need to be decipherable.

Response: Thank you for your professional comments. We revised Figure 4 to make the word and cis-regulatory elements clearer.

Comment 4: Figure 6 is another important Figure that is difficult to read. Please enlarge and revise accordingly.

Response: We revised Figure 6 according to both you and other reviewers.

Comment 5: The Discussion section needs some revision. There are long sentences and long paragraphs, and the text would benefit greatly from some reorganization.

Response: We changed the long sentences in “Discussion” to short sentences. Please see the yellow highlighted sentences in the discussion.

Reviewer 3 Report

Comments and Suggestions for Authors

Thank you for your carefully prepared manuscript. While the study is well-structured, I have several minor suggestions to enhance readability:

  1. Italicize gene names (WOX) throughout the text (e.g., Lines 48, 86, 98, 113, 144, 218, 451).
  2. For A. thalianaAt (not AraWOXor ArtWOX) is the established abbreviation (used in most publications). For Amborella trichopodaAmtrWOX is recommended (https://www.nature.com/articles/s41598-025-88880-x).
  3. Correct typos: "WXO" → WOX (Lines 179, 374).
  4. Increase resolution and size of Figure 2a,b, Figures 4 and 6.
  5. Specify the software used to generate the heatmap circle.
  6. For qRT-PCR data (Figure 5), ANOVA (or a similar post-hoc test) should replace the t-test to account for multiple comparisons.

Author Response

Comment 1: Italicize gene names (WOX) throughout the text (e.g., Lines 48, 86, 98, 113, 144, 218, 451).

Response: We italicize gene names throughout our manuscript according to your suggestion.

Comment 2: For A. thalianaAt (not AraWOXor ArtWOX) is the established abbreviation (used in most publications). For Amborella trichopodaAmtrWOX is recommended (https://www.nature.com/articles/s41598-025-88880-x).

Response: Thank you for your professional comments. We changed the abbreviations as you suggested.

Comment 3:  Correct typos: "WXO" → WOX (Lines 179, 374).

Response: We revised this error in lines 178, 377.

Comment 4:  Increase resolution and size of Figure 2a,b, Figures 4 and 6.

Response: We increase the resolution of all figures to 600 pixels and submit them individually.

Comment 5: Specify the software used to generate the heatmap circle.

Response: We add the method “circlize R package” in line 473.

Comment 6: For qRT-PCR data (Figure 5), ANOVA (or a similar post-hoc test) should replace the t-test to account for multiple comparisons.

Response: According to your suggestion. The t-test was replaced by the ANOVA test in Figure 5.

Reviewer 4 Report

Comments and Suggestions for Authors

Review Plants 3662070

In their manuscript titled “Genome-wide identification and expression analysis of the WUSCHEL family of homeobox genes in Nelumbo evolution,” Li and Zhang identified members of the WOX family of transcription factors in yellow lotus (Nelumbo lutea) and analyzed their evolutionary patterns by studying the species divergence of N. lutea and its related species N. nucifera. Using bioinformatic programs, the authors revealed 11 NlWOX genes and categorized them into three clades. They presented data on the chromosomal location of NlWOX genes and gene duplications in the N. lutea genome, as well as on the collinearity of NlWOX genes between N. lutea and N. nucifera. Particular attention is paid to tissue expression patterns of NlWOX and cis-regulatory elements in WOX promoters.  Based on conserved motif and protein sequence structures of WOX, the authors conclude that the two genomes of the lotus have undergone variation in specific regions while also retained some conserved collinear regions. Overall,  these  data provide valuable insights into the evolution and function of the WOX family in  the genus Nelumbo

In addition,“To verify the tissue expression patterns of the NlWOX genes” ,Li and Zhang presented the results of RT PCR analysis of six expressed WOX genes in apical meristem and various organs which they claim “were consistent with the RNA-seq sequencing, demonstrating the accuracy of our results “(line 271). However, no data on RNA-seq sequencing is depicted in Figure 5 in accord with the usual procedure of validation the stringency of RNA-seq tests. Furthermore, the accuracy of the co-expression networks for NnWOX genes, which were constructed ““based on the pipeline of weighted gene co-expression network analysis (WGCNA) R-package the WGCNA networks with a min-module of 500 genes “, as well as the GO enrichment analysis of the co-expressed NlWOX genes, is difficult to assess because the authors did not provide supplementary materials on the MDPI website, while the supporting information that is supposedly downloadable at: 510 www.mdpi.com/xxx/s1 returns a 404 error.

(Additional feedback after received supplementary materials)

Although the authors presented a Heatmap that  shows the correlations between WGCNA gene modules and tissues, the exact genes that are coregulated  with WOX family genes and that  form fourteen “color modules” are left beyond the frames of Supplementary  materials. I strongly recommend that the authors elaborate on the description of the major groups of genes that form the “minimal 500-gene module.” Otherwise, the authors' claims that WOX genes were involved in different biological pathways between the two lotus species remain conceptual and to some extent illusory.

The article would undoubtedly be of greater interest to readers if the authors had covered in detail the genes coregulated with the studied transcriptional factors and the nature of the differences between lotus species, without reducing them to “color modules”

Author Response

Comment 1:In addition,“To verify the tissue expression patterns of the NlWOX genes” ,Li and Zhang presented the results of RT PCR analysis of six expressed WOX genes in apical meristem and various organs which they claim “were consistent with the RNA-seq sequencing, demonstrating the accuracy of our results “(line 271). However, no data on RNA-seq sequencing is depicted in Figure 5 in accord with the usual procedure of validation the stringency of RNA-seq tests. Furthermore, the accuracy of the co-expression networks for NnWOX genes, which were constructed ““based on the pipeline of weighted gene co-expression network analysis (WGCNA) R-package the WGCNA networks with a min-module of 500 genes “, as well as the GO enrichment analysis of the co-expressed NlWOX genes, is difficult to assess because the authors did not provide supplementary materials on the MDPI website, while the supporting information that is supposedly downloadable at: 510 www.mdpi.com/xxx/s1 returns a 404 error.

Response: Thank you for your professional comments. The tissue expression patterns of NlWOX genes from RNA-seq were shown in Figure 4a. Therefore, we need not to show them again in Figure 5. To more accurately present the co-expressed genes of NlWOX, we have added these co-expression relationships and their correlation coefficients to the revised Table S1.

Comment2: Although the authors presented a Heatmap that  shows the correlations between WGCNA gene modules and tissues, the exact genes that are coregulated  with WOX family genes and that  form fourteen “color modules” are left beyond the frames of Supplementary  materials. I strongly recommend that the authors elaborate on the description of the major groups of genes that form the “minimal 500-gene module.” Otherwise, the authors' claims that WOX genes were involved in different biological pathways between the two lotus species remain conceptual and to some extent illusory. The article would undoubtedly be of greater interest to readers if the authors had covered in detail the genes coregulated with the studied transcriptional factors and the nature of the differences between lotus species, without reducing them to “color modules”

Response: According to your comments, we have added these co-expression relationships and their correlation coefficients to the revised Table S1. Additionally, the color modules in our WGCNA analysis were related to the tissues. Based on the correlations between these color modules and tissues, we can associate these WOX genes with specific tissues. By comparing the tissue correlations of WOX co-expression networks between the two species, we analyzed the expression divergence of this gene family during species evolution. Given the variations in enriched functions of co-expressed genes among collinear genes of the two species, we have reason to believe that variations in tissue expression patterns promoted changes in the biological functions of WOX between the two species.

Round 2

Reviewer 2 Report

Comments and Suggestions for Authors

Authors have revised the manuscript and responded to my comments. 

The manuscript can now be published.

Author Response

Comment1:Authors have revised the manuscript and responded to my comments. 

The manuscript can now be published.

Response:  Thank you for your recognition.

Reviewer 4 Report

Comments and Suggestions for Authors

Review 2 Plants 3662070

Unfortunately, I could not see any changes in the manuscript according to the expressed  remarks.  The authors did not change Figure 5, referring to the fact that the transcriptomic data are presented in revised Table S1T. It is rather inconvenient for the readers to compare the table and figure and they, therefore, have to take the authors' statements about the correspondence between the PCR results and the transcriptomic data on faith.

Revised Table S1 to which the authors refer in response to the second remark is not available (FILE ILLEGAL), and there are no comments in the text about the nature of co-expressed genes. I do not dispute the authors' assertions that “Given the variations in enriched functions of co-expressed genes among collinear genes of the two species, we have reason to believe that variations in tissue expression patterns promoted changes in the biological functions of WOX between the two species”. However, I, as well as the readers, would like to understand the essence of the phenomena underlying the tissue specificity of co-expressed genes.

Author Response

Comment1: Unfortunately, I could not see any changes in the manuscript according to the expressed  remarks.  The authors did not change Figure 5, referring to the fact that the transcriptomic data are presented in revised Table S1T. It is rather inconvenient for the readers to compare the table and figure and they, therefore, have to take the authors' statements about the correspondence between the PCR results and the transcriptomic data on faith.

Response: According to your suggestions, I have revised Figure 5 to display the results of qRT-PCR and RNA-seq in the sam bar chart. Our results clearly demonstrate their consistency.

Comment2: Revised Table S1 to which the authors refer in response to the second remark is not available (FILE ILLEGAL), and there are no comments in the text about the nature of co-expressed genes. I do not dispute the authors' assertions that “Given the variations in enriched functions of co-expressed genes among collinear genes of the two species, we have reason to believe that variations in tissue expression patterns promoted changes in the biological functions of WOX between the two species”. However, I, as well as the readers, would like to understand the essence of the phenomena underlying the tissue specificity of co-expressed genes.

Response: We have uploaded Supplementary Table 1. Meanwhile, in the Discussion section, we added speculations on the expression divergence of these collinear genes (lines 397-398). We also interpreted the biological significance represented by the enrichment analysis of these co-expressed gene differences (lines 399).

Round 3

Reviewer 4 Report

Comments and Suggestions for Authors

Review3

The authors  have revised Figure 5 to display the results of qRT-PCR and RNA-seq in the same bar chart.

However,  Supplementary Table1 is still unavailable which does not allow us to draw any conclusions about the genes co-expressed with WOX genes. I don't mind that that”The enriched biological functions of co-expressed genes for NnWOXs and NlWOXs diverged, suggesting that they participated in different pathways”. However, I and the readers would like to know in which ones. Without this information, the work is reduced to a simple analysis of the evolution of the WOX gene family without understanding the significance of these transcription factors for the biology of the genus or their use in practical breeding.

Author Response

Comments:However,  Supplementary Table1 is still unavailable which does not allow us to draw any conclusions about the genes co-expressed with WOX genes. I don't mind that that”The enriched biological functions of co-expressed genes for NnWOXs and NlWOXs diverged, suggesting that they participated in different pathways”. However, I and the readers would like to know in which ones. Without this information, the work is reduced to a simple analysis of the evolution of the WOX gene family without understanding the significance of these transcription factors for the biology of the genus or their use in practical breeding.

Response: Thank you for your comments. I have sent the attached file to you separately, which shows the co-expressed genes of NLWOX. Regarding your question, we kindly request you to compare the enrichment analysis of NnWOX co-expressed genes in our previous study with that in this paper (doi: 10.3390/plants13050720.). The core of this article is to analyze the divergence of WOX gene expression patterns between the two species. Whether the co-expression relationships represent accurate regulatory relationships requires further experimental validation, which is also the next step of our research plan.

Round 4

Reviewer 4 Report

Comments and Suggestions for Authors

The functions of the co-expressed genes in the attached file, which the authors kindly sent separately, still remain enigmatic, despite some very sparse comments on the enrichment analysis of these genes. However  given the authors’ aim to analyze “the divergence of WOX gene expression patterns between the two species”, I can only wish them success in their further research of  functional relationships between  WOX family and co-expressed genes.